# Universal Likelihood Rewards
# for LLM Reasoning

## Abstract

Fine-tuning large language models (LLMs) for reasoning is usually done by reinforcement learning on reasoning benchmarks, with a specific reward function, often binary, for each benchmark. Here, we systematically investigate chain-of-thought (CoT) training in LLMs using rewards derived from the probability or log-probability of emitting the reference answer (or any other prompt continuation present in the data). Several recent works have advocated for the use of similar rewards (e.g., VeriFree, JEPO, RLPR, NOVER), which have the advantage of not relying on specific verifiers and being available at scale.

We systematically compare such probability-based variants with standard baselines, testing performance both on standard mathematical reasoning benchmarks, and on long-form answers where no external verifier is available. We find that using the *log-probability* of the reference answer as the reward for CoT learning is the only option that performs well in all setups. This is also consistent with the next-token log-likelihood loss used during pretraining.

In verifiable settings, log-probability rewards bring comparable or better success rates than reinforcing with standard binary rewards, and yield much better perplexity. In non-verifiable settings, they perform on par with SFT. On the other hand, methods based on probability, such as VeriFree, flatline on non-verifiable settings due to vanishing probabilities of getting the correct answer.

Overall, this establishes log-probability rewards as a viable method for CoT fine-tuning, bridging the short, verifiable and long, non-verifiable answer settings.

## 1 Introduction

Large language models (LLMs) have achieved striking progress on tasks requiring reasoning, from mathematics to code generation (Cobbe et al., 2021; Hendrycks et al., 2021b; OpenAI, 2023). A central ingredient has been chain-of-thought (CoT) prompting, where models articulate intermediate reasoning steps before producing a final answer (Wei et al., 2022; Guo et al., 2025). However, CoTs are rarely available in raw training data, making reinforcement learning (RL) the predominant approach: the CoT is treated as a sequence of actions, and correctness of the final answer determines the reward. This paradigm works well in verifiable domains such as mathematics and programming, where ground-truth correctness is available (Cobbe et al., 2021; Hendrycks et al., 2021b; Chen et al., 2021; Austin et al., 2021; Hendrycks et al., 2021a), but it does not naturally extend to non-verifiable domains like long-form proofs or open-ended generation.

To overcome this limitation, we investigate reusing training signals closer to the log-likelihood signal already employed during pretraining. Instead of sampling answers and relying on 0/1 correctness rewards, we reward the model for increasing the probability or log-probability of the answers present in the training data. Such criteria are universal—they apply in both verifiable and non-verifiable settings and could provide a denser signal. Such approaches are present, e.g., in Zhou et al. (2025) (training with the probability of the reference answer) or Tang et al. (2025) (training with a variant of the log-probability of the reference answer). Note that reference answers are available for situations in which 0/1 rewards are not, such as long-form question-answering. This makes it possible to test these methods both in verifiable and nonverifiable, long-form answer settings. We particulary focus on the case of log-probability, since it is conceptually the closest to the pretraining criterion.

**Our Approach and Contributions.** We conduct the first comprehensive study of probability-based RL rewards for CoT training, spanning verifiable and non-verifiable domains, across multiple model families (Qwen-2.5, Llama-3.2). Our main contributions and findings are:

- **Systematic evaluation across domains.** We test many variants of probability-based rewards (probabilities and log-probabilities, including several variants from the literature such as VeriFree, RLPR, JEPO) for CoT training, comparing against supervised fine-tuning (SFT) and standard RL training (RLOO) baselines. We run the comparisons on two verifiable benchmarks – MATH (Hendrycks et al., 2021b), DeepScaleR (Luo et al., 2025) – and two non-verifiable settings - Alpaca (Taori et al., 2023) and the non-verifiable "proof portion" of NuminaMath (Li et al., 2024).

- **Universality of log-probability rewards.** Among the variants tested, rewards based on *log*-probabilities perform well in every scenario (short, verifiable answers and long, non-verifiable answers), while all others fail in one or several settings.

- **Advantages of probability-based rewards.** For verifiable domains, all variants of probability-based rewards perform similarly and slightly outperform base RL training in terms of greedy success rate on verifiable domains. They also offer some computational advantages during training (no need to sample an answer).

- **Success and perplexity trade-offs.** In verifiable domains, log-probability rewards perform well both in terms of success rate and of perplexity – a key metric aligned with pretraining. On the other hand, both base RL training and probability-based rewards perform extremely poorly on perplexity. This highlights a distinct advantage of log-probabilites.

- **Non-verifiable domain behavior.** On long-form domains, both base RL and pure probability rewards collapse due to vanishing probabilities of long answers. Log-probability rewards remain viable and perform similarly to SFT.

- **CoT shortening with log-probability rewards.** In every scenario, log-probability rewards lead to an initial shortening of the CoT. For verifiable domains, the length of the CoT recovers during training. On the other hand, for non-verifiable domains, the CoT stays very short, meaning log-probability rewards largely follow SFT from that point. On verifiable domains, base RL and pure probability rewards (VeriFree) do not exhibit this shortening. Mitigating strategies such as CoT length rewards and KL penalties maintain CoT but hurt performance. Thus, it seems that RL CoT training on nonverifiable domains can only match SFT by eliminating the CoT. We discuss hypotheses around this phenomenon.

Overall, these results establish log-likelihood rewards as a simple way to bridge verifiable and non-verifiable settings under a single training criterion, broadly applicable for fine-tuning LLMs.

**Related Work.** Several prior works have proposed to modify the binary rewards in standard RL post-training settings. We can globally distinguish these rewards into *intrinsic* rewards that do not require ground-truth, and those that use the *confidence or log-likelihood of the ground-truth* answer. The former category utilizes measures of confidence, entropy or diversity as measured by the generating language model itself (Prabhudesai et al., 2025; Agarwal et al., 2025; Zhao et al., 2025; Li et al., 2025a; Gao et al., 2025). Nevertheless, these intrinsic rewards generally cannot surpass rewards grounded in true correctness except under strong coverage assumptions, and tend to lead to reward hacking or diversity collapse. Huang et al. (2025a) show that self-rewarding can only "sharpen" knowledge already covered by the base model—it cannot create new information—so performance is bounded by model coverage (i.e. its pass@k rate). Song et al. (2024) formalizes the generation–verification gap and shows self-improvement hinges on sufficient coverage and verifier quality; when these are weak, intrinsic/self-verification stalls and fails to match correctness-based training. Finally, Huang et al. (2025b) proves that inference-time alignment with imperfect reward models suffers reward hacking and lacks guarantees under realistic coverage, again falling short of what verified rewards can achieve. Another study (sur Kayal et al., 2025), shows that certain intrinsic signals (like policy entropy or state novelty) can fail in high-dimensional or complex output spaces, and sometimes result in exploration that diverges from the downstream task. Some works combine intrinsic and binary rewards (Song et al., 2025; Li et al., 2025b) to encourage exploration. Yet another line of works explores using LLM-as-a-judge synthetic rewards in RL-based post-training (RLAIF), explored as an alternative to human feedback (Lee et al., 2024; Bai et al., 2022) or for (semi-)verifiable domains (Whitehouse et al., 2025; Jayalath et al., 2025; Simonds et al., 2025).

Closer to our line of work are works that use the probability or log-likelihood of the reference answer given a generated reasoning chain under the initial policy model to provide a verifier-free scoring function. We highlight the works relevant to our setting and label them with distinctive keywords for clarity. To the best of our knowledge, none of these studies investigate log-likelihoods as a primary reward signal, with the exception of Tang et al. (2025), who include it as an ablation against their proposed JEPO reward and report weaker performance. In contrast, we introduce log-likelihood rewards as a primary training signal, and our experiments consistently demonstrate their competitiveness across models and datasets, including those evaluated in prior work.

- *VeriFree* (Zhou et al., 2025) uses probabilities of reference answers as reward in verifiable domains
- *JEPO* (Tang et al., 2025) introduces a Jensen based ELBO loss with log-probs. In experiments they mix verifiable with non-verifiable data to show that the verifiable part improves with this loss.
- *RLPR* (Yu et al., 2025) uses average probability of the ground truth for non-verifiable domains.
- *NOVER* (Liu et al., 2025) is a variant of probability-based rewards, using a geometric mean of per-token perplexities.
- *Reinforcement-pretraining* (Dong et al., 2025) performs small-scale *pretraining* from scratch, inserting CoTs at specific points and using logprob rewards.
- *LongForm* (Gurung & Lapata, 2025) designs a clever reward function (VR-CLI) that allows them to use an unlabeled book dataset as a learning signal for reasoning.

## 2 METHOD

**Context: Chain-of-thought fine-tuning via Reinforcement Learning.** We consider the general context of fine-tuning an LLM to improve performance on a set of questions-answers via a Chain-of-Tought (CoT) optimized by reinforcement learning. For each prompt $p$, the fine-tuned model should first print a CoT $z$, then an answer $a$. Then a reward $R$ is computed depending on $a$ (such as correctness, or matching some reference answer). Fine-tuning should optimize the expected reward.

Denoting $\pi_\theta$ the generative probabilistic model with parameter $\theta$, and $\mathcal{D}$ the dataset (a distribution of questions or prompts $p$), we want to maximize

$$J_\theta = \mathbb{E}_{p \sim \mathcal{D}} \mathbb{E}_{z \sim \pi_\theta(z|p), \, a \sim \pi_\theta(a|p,z)}[R(z, a)] \tag{1}$$

where $R(z, a)$ is the reward obtained for CoT $z$ and answer $a$.

This task is often tackled with RL variants of the basic Reinforce algorithms, such as RLOO (Ahmadian et al., 2024), GRPO (Guo et al., 2025), or PPO (Schulman et al., 2017).

**RL fine-tuning with probability-based rewards.** We focus on the case when a reference answer $a^\star$ is available for each prompt in the dataset. Then it is possible to estimate the probability of this answer given the CoT. We will compare RL training with several rewards derived in this setting.

For instance, we can set a reward similar to the log-loss used during pretraining,

$$R(z, a) = \log \pi_\theta(a^\star|p, z). \tag{2}$$

We call this setting *log-prob rewards*. Given a CoT $z$, this quantity can be computed in one pass of a transformer on the reference answer $a^*$. In particular, since the reward depends on $z$ but not on $a$, sampling of an answer $a$ given the CoT $z$ is not necessary.

We also consider the *average log-prob reward* variant

$$R(z, a) = \frac{1}{|a^\star|} \log \pi_\theta(a^\star|p, z) \tag{3}$$

namely, we compute the per-token log-probability by downscaling the reward by the length $|a^\star|$ of the answer. This results in a different weighting of the various data samples in the dataset.

Log-prob rewards are aligned with the pretraining phase of LLM training, where the criterion is the log-probability of the next token. They do not require access to a verifier, only to a reference answer (or any continuation) in the data. Thus, they can potentially be applied any question-answer pairs.

The logprob reward setting is also considered in Tang et al. (2025), although they largely focus on a "multi-sample" variant. The gradient of the expected reward is derived there as

$$\nabla J_\theta = \mathbb{E}_{p \sim \mathcal{D}} \, \mathbb{E}_{z \sim \pi_\theta(z|p), \, a \sim \pi_\theta(a|p,z)}$$
$$[\log \pi_\theta(a^\star|p,z) \nabla \log \pi_\theta(z|p) + \nabla \log \pi_\theta(a^\star|p,z)] \quad (4)$$

As noted in Tang et al. (2025), the second term is analogous to a supervised fine-tuning term that directly optimizes the log-likelihood of the reference answer $a^\star$ given what comes before, and the first term is a traditional Reinforce term with reward $\log \pi_\theta(a^\star|p,z)$. For completeness, we derive this gradient in Appendix A, together with its application to RL algorithms such as RLOO.

A related but different reward appears in Zhou et al. (2025):

$$R_{\text{VeriFree}}(z, a) = \pi_\theta(a^\star|p, z) = \mathbb{E}_{a \sim \pi_\theta(a|p,z)}[1_{a=a^*}] \quad (5)$$

thus, without the logarithm. This is the *expected* success rate for matching the reference answer $a^*$: *in expectation*, it is the same as sampling an answer $a$ given the CoT, and setting a reward 1 if $a = a^*$. Zhou et al. (2025) prove that working with the expectation reduces variance compared to sampling $a$, and this affects training dynamics.

The VeriFree reward diverges from logprob rewards when probabilities are very small. For instance, if initially the model has an almost-zero probability to reach the reference answer, then the VeriFree reward produces no learning. Similarly, for long free-form answers, the probability of an exact match with $a^*$ is tiny, so we would expect a difference between VeriFree and logprob rewards. On the other hand, if the initial probability to reach the correct answer is reasonably high, then we expect the VeriFree and logprob rewards to be well aligned.

**Algorithms tested.** We now give an outline of the algorithms compared in the experiments.

For every algorithm except JEPO, the advantages used for the Reinforce gradient updates are obtained by RLOO, i.e., by subtracting from the reward a leave-one-out estimate of the mean reward estimated on a minibatch for a given prompt; this is an unbiased version of GRPO (Guo et al., 2025).

- *SFT*: standard fine-tuning with the next-token cross-entropy loss. Namely, we omit the CoT, and fine-tune the model to predict the ground truth directly from the prompt.
- *Base RL*: this is the most direct RL method. For each prompt $p$, we sample a CoT $z \sim \pi_\theta(z|p)$, then an answer $a \sim \pi_\theta(a|p,z)$, and check whether the answer is correct:

$$R_{\text{RLOO}}(z, a) = 1_{a=a^\star} \quad (6)$$

  given the reference answer $a^\star$. As for all other RL methods, we employ a leave-one-out advantage estimation (RLOO).
- *Probability* (VeriFree): As mentioned above, the reward is

$$R_{\text{Probability}}(z, a) = \pi_\theta(a^\star|p, z) = \mathbb{E}_{a \sim \pi_\theta(a|p,z)}[1_{a=a^*}] \quad (7)$$

  namely, instead of sampling an answer $a$ from the model, we directly compute the probability of the reference answer $a^*$ given $z$ using the model $\pi_\theta$.
- *Average prob* (AvgProb): Similarly to RLPR (Yu et al., 2025), the reward is set to the *average per-token probabilities* of the reference answer:

$$R_{\text{avgprob}}(z, a) = \frac{1}{|a^\star|} \sum_{t=1}^{|a^\star|} \pi_\theta(a_t^\star|p, z, a_{[1:t-1]}^\star) \quad (8)$$

- *Log-prob*: the reward is

$$R_{\text{log-prob}}(z, a) = \log \pi_\theta(a^\star|p, z) \quad (9)$$

  namely, we directly compute the log-likelihood of the reference answer $a^*$ given $z$.
- *Average log-prob* (AvgLogprob): In log-probs, longer answers have rewards of a bigger magnitude, since $\log \pi_\theta(a^\star|p, z)$ is a sum over all tokens in $a^\star$. Average log-probs rescales the reward accordingly:

$$R_{\text{avglogprob}}(z, a) = \frac{1}{|a^\star|} \log \pi_\theta(a^\star|p, z) \quad (10)$$

  where $|a^\star|$ is the number of tokens in $a^\star$. Compared to log-probs, this just means that different answers in the dataset are weighted in a different way.

- *JEPO* (Tang et al., 2025) used a refined version of the group reward in GRPO and RLOO, by noting that the expected log-probability $\mathbb{E}_{z \sim \pi_\theta(z|p)} \log \pi_\theta(a^\star|p, z)$ is an underestimate of the actual log of the probability to get $a^\star$ using $\pi_\theta$, which is $\log \mathbb{E}_{z \sim \pi_\theta(z|p)} \pi_\theta(a^\star|p, z)$. So, starting from GRPO, they introduce a group-level reward based on $G$ samples $z_1, \ldots, z_G$ for a given prompt,

$$R(z_1, \ldots, z_G) = \log \frac{1}{G} \sum_{i=1}^{G} \pi_\theta(a^\star|p, z_i). \tag{11}$$

Compared to log-probs over a similar minibatch $z_i$, the reward is the log-mean-exp of rewards in the minibatch. For Reinforce advantage estimation, they subtract the similar estimate over $G - 1$ samples without the sample $z_i$. We will use $G = 4$ as in Tang et al. (2025).

**Success metrics.** For each algorithm, we report several success metrics. These metrics largely follow the quantities tracked by the different algorithms.

We denote by $\mathcal{D}$ the distribution of prompts and reference answers in the dataset.

Given a prompt $p$, the probability to obtain the correct answer using a CoT model $\pi$ is

$$\pi^{\text{CoT}}(a^\star|p) = \mathbb{E}_{z \sim \pi(z|p)} \left[ \pi(a^\star|p, z) \right]. \tag{12}$$

- *Success rate*: This is the probability to get a correct answer, averaged over the dataset,

$$\mathbb{E}_{(p,a^\star) \sim \mathcal{D}} \left[ \pi^{\text{CoT}}(a^\star|p, z) \right]. \tag{13}$$

It can be estimated directly by sampling a prompt and answer in the dataset, sampling a CoT $z$, and computing $\pi_\theta(a^\star|p, z)$. This is the estimate we report. VeriFree and Direct RLOO directly optimize the success rate. We consider two modes for generating the answers given a prompt and chain of thought: *Greedy success*, where the most likely token is used at each step, and $T = 1$ *sampling success* from the softmax probabilities at temperature $T = 1$.

- *Log-probability*: This is a family of metrics that aggregate the likelihood of answer tokens across the dataset.

$$\mathbb{E}_{(p,a^\star) \sim \mathcal{D}} \left[ \log \pi^{\text{CoT}}(a^\star|p, z) \right]. \tag{14}$$

To keep these quantities comparable, we consider two averaging schemes - *per-token* and *per-answer*.

  - *Per-token log-probabilities* sums the log-probabilities of all answer tokens in the dataset, and divides by the total number of those tokens. Equivalently

$$\frac{1}{\mathbb{E}_{(p,a^\star) \in \mathcal{D}}[|a^\star|]} \mathbb{E}_{(p,a^\star) \in \mathcal{D}} \left[ \log \pi^{\text{CoT}}(a^\star|p) \right] \tag{15}$$

  - *Per-answer log-probabilities* averages across each answer, then averages over the dataset:

$$\mathbb{E}_{(p,a^\star) \in \mathcal{D}} \left[ \frac{1}{|a^\star|} \log \pi^{\text{CoT}}(a^\star|p) \right] \tag{16}$$

These also correspond to the quantities optimized by the Log-probs and Average log-probs methods, respectively.

Note however that this is difficult to estimate directly due to the expectation over $z$ inside the $\log$, since $\pi^{\text{CoT}}$ is an expectation. A simple solution is to estimate the average via Monte Carlo using $N$ samples (Tang et al., 2025):

$$\mathbb{E}_{(p,a^\star) \sim \mathcal{D}} \left[ \log \frac{1}{N} \sum_{i=1}^{N} \pi(a^\star|p, z_i) \right] \tag{17}$$

where the $z_i$ are sampled from $\pi(z_i|p)$. We refer to this as *logprob-MCN*. We apply this modification to both per-answer and per-token averaged logprobs.

We will use both the "naive" estimate logprob-MC1 with $N = 1$, and a more precise estimate, *logprob-MC32*, computed less frequently during training.

For supervised fine-tuning (SFT) with no CoT, this is irrelevant as there is no expectation over $z$, and we can report $\log \pi(a^\star|p)$ directly.

The MC estimate is always an *underestimate* of the actual logprob $\log \pi^{\text{CoT}}(a^\star|p, z)$, since $\log$ is concave. This is especially relevant when comparing logprob-MC1 to SFT log-probabilities.

- We also report *perplexity*, which is just the exponential of minus per-answer log-probabilities. This corresponds to the geometric mean over the dataset, of the perplexity of the answer for each prompt. Technically, this corresponds to *per answer perplexity-MC1*; we shorten to *perplexity*.

- *Average CoT length*: We also report the average length of the CoTs used by a model,

$$\mathbb{E}_{(p,a^\star)\in\mathcal{D}}\,\mathbb{E}_{z\sim\pi(z|p)}\left[|z|\right] \tag{18}$$

as a relevant quantity for analysis. Note that this includes formatting tokens.

## 3 EXPERIMENTAL RESULTS

### 3.1 SETUP: DATASETS, MODELS, AND PROTOCOL

**Models.** We evaluate on two instruction-tuned models: LLAMA-3.2-3B-INSTRUCT (Dubey et al., 2024), and QWEN-2.5-3B-INSTRUCT (Yang et al., 2024).

**Datasets.** We consider two *verifiable* math benchmarks and two *non-verifiable* long-form datasets. (i) **MATH** (Hendrycks et al., 2021b):We report accuracy on the official test split. The resulting training set contains $\sim$7,000 short-answer problems. (ii) **DeepScaleR (Preview)** (Luo et al., 2025): we hold out a random $10\%$ for validation to report performance. The training set has $\sim$39,000 short-answer problems. (iii) **Alpaca (cleaned)** (Taori et al., 2023): we use the standard cleaned variant; 1,000 random examples are used for validation, leaving $\sim$50,000 training samples with predominantly long-form answers. (iv) **NuminaProof**: starting from NUMINAMATH-1.5 (Li et al., 2024), we filter for theorem–proof style items. We reserve 1,000 examples for validation, yielding $\sim$50,000 long-form training samples. Mote detail in Appendix B.

**Algorithms tested.** We compare the algorithms mentioned in Section 2, namely, SFT and the following RL variants: Base RL, Probability (VeriFree), Logprob, AvgLogprob, AvgProb, and JEPO. These differ by the rewards used, as described in Section 2. Details in Appendix B.

*Verifiable.* We run experiments with all methods on verifiable domains with $G = 4$ and $G = 32$, except for JEPO, where we only run with group size $G = 4$ (the value used in Tang et al. (2025)) as JEPO is harder to implement efficiently for larger $G^1$. In the loss function we include a KL divergence regularization term as proposed by Guo et al. (2025) with a coefficient of 0.001.

*Non-verifiable.* In non-verifiable domains, we run with $G = 4$ throughout. Here, we do not use a KL divergence term in the main results, but we explore its impact on CoT-length and performance in the ablations.

### 3.2 RESULTS ON VERIFIABLE DOMAINS

We present the results on verifiable domains in Table 1, and Figures 1 and 5 to 7 in Appendix C for $G = 32$ and in Table 3 and Figures 8 to 11 in Appendix C for $G = 4$, the latter including JEPO. The key takeaway is that *all RL variants based on ground-truth answers perform comparably* for greedily decoded answers. More precisely, all (log-) probability-based variants perform better than Base RL when run with standard group size $G = 32$.

Sampling answers at temperature $T = 1$ generally makes performance worse across the board. It also affects the ranking of methods: methods that use logprobs or average logprobs underperform both Base RL and the Prob variant. We believe $T = 1$ sampling is the reason why logprobs did not perform well on MATH in Tang et al. (2025). Overall, we do not detect any strong difference between JEPO and simple Logprob when greedy sampling is used. Conceptually, JEPO is a more precise, more computationally heavy version of Logprob (larger $N$ for Monte Carlo estimation of log-probabilities, see Section 2). The additional complexity is not justified in our setting.

The picture shifts when we consider perplexity: here, only Logprob, AvgLogprob and JEPO achieve good perplexities. Perplexity may not be the metric of most direct interest for verifiable questions, but it nevertheless informs us on the qualitative behavior of different models. For instance, a prob-trained model makes little difference between predicting a wrong answer with probability 0.99 or

---

[1] Because the JEPO reward depends on the whole group and cannot be computed for each sample independently, efficient implementation with large $G$ is more delicate.

|  | Base model | Base RL | Log-prob | Avg Logprob | Probability | Avg Probability | SFT (no CoT) |
|---|---|---|---|---|---|---|---|
| **Llama 3B, MATH** | | | | | | | |
| Greedy success | 17.13 ± 0.00 | 42.74 ± 0.66 | **43.30 ± 0.10** | 43.66 ± 0.25 | **44.19 ± 0.57** | 43.95 ± 0.44 | 14.43 |
| T=1 Sampled Success | 10.77 ± 0.27 | **41.36 ± 0.15** | 34.66 ± 0.81 | 34.83 ± 0.00 | **41.39 ± 0.09** | 38.38 ± 0.74 | 10.00 |
| Per-answer avg logprob MC32 | — | — | **-0.68 ± 0.00** | -0.69 ± 0.01 | -1.32 ± 0.01 | -0.77 ± 0.00 | -0.97 |
| Per-answer avg logprob | -4.21 ± 0.00 | -2.63 ± 0.02 | **-0.79 ± 0.03** | -0.81 ± 0.04 | -1.97 ± 0.04 | -1.08 ± 0.03 | -0.97 |
| Perplexity | 67.29 ± 0.00 | 13.87 ± 0.34 | **2.21 ± 0.06** | 2.25 ± 0.09 | 7.14 ± 0.26 | 2.95 ± 0.08 | 2.63 |
| CoT length | 326.77 ± 0.71 | 321.01 ± 24.42 | 298.97 ± 2.82 | 302.74 ± 2.21 | 313.58 ± 1.87 | 320.17 ± 5.69 | 5.00 |
| **Qwen 3B, MATH** | | | | | | | |
| Greedy success | 21.19 ± 0.00 | 55.85 ± 0.46 | 56.84 ± 0.21 | 56.11 ± 0.20 | 56.46 ± 0.05 | **57.41 ± 0.16** | 18.32 |
| T=1 Sampled Success | 16.63 ± 0.30 | **55.36 ± 0.17** | 44.18 ± 0.42 | 42.45 ± 0.51 | 53.32 ± 0.22 | 47.34 ± 0.87 | 12.00 |
| Per-answer avg logprob MC32 | — | — | **-0.39 ± 0.00** | -0.40 ± 0.01 | -1.03 ± 0.01 | -0.42 ± 0.00 | -0.69 |
| Per-answer avg logprob | -2.19 ± 0.00 | -2.11 ± 0.10 | **-0.44 ± 0.02** | -0.50 ± 0.00 | -1.77 ± 0.01 | -0.68 ± 0.05 | -0.69 |
| Perplexity | 8.92 ± 0.00 | 8.25 ± 0.80 | **1.55 ± 0.03** | 1.64 ± 0.00 | 5.85 ± 0.03 | 1.97 ± 0.09 | 1.99 |
| CoT length | 222.35 ± 0.73 | 451.83 ± 14.14 | 381.49 ± 10.22 | 372.01 ± 18.76 | 429.35 ± 1.25 | 419.51 ± 4.31 | 5.00 |
| **Llama 3B, DeepScaleR** | | | | | | | |
| Greedy success | 10.05 ± 0.00 | 25.45 ± 1.06 | **30.60 ± 2.01** | 28.73 ± 1.13 | 29.03 ± 0.88 | **29.60 ± 1.91** | 8.43 |
| T=1 Sampled Success | 5.95 ± 0.07 | **25.46 ± 0.69** | 15.17 ± 1.96 | 15.77 ± 0.75 | **25.19 ± 0.97** | 19.62 ± 1.44 | 4.27 |
| Per-answer avg logprob MC32 | — | — | **-0.85 ± 0.02** | -0.85 ± 0.02 | -1.77 ± 0.18 | -0.90 ± 0.00 | -1.07 |
| Per-answer avg logprob | -4.92 ± 0.00 | -3.18 ± 0.07 | **-0.94 ± 0.03** | -1.02 ± 0.12 | -2.41 ± 0.19 | -1.17 ± 0.03 | -1.07 |
| Perplexity | 137.25 ± 0.00 | 24.01 ± 1.73 | **2.56 ± 0.08** | 2.77 ± 0.34 | 11.28 ± 2.13 | 3.22 ± 0.09 | 2.92 |
| CoT length | 345.75 ± 15.40 | 266.94 ± 59.84 | 321.53 ± 11.91 | 346.77 ± 16.29 | 343.26 ± 48.40 | 350.66 ± 7.19 | 5.00 |
| **Qwen 3B, DeepScaleR** | | | | | | | |
| Greedy success | 14.02 ± 0.00 | **37.02 ± 0.46** | 35.38 ± 0.11 | 36.93 ± 0.85 | 37.35 ± 0.35 | 37.59 ± 1.22 | 9.96 |
| T=1 Sampled Success | 9.55 ± 0.07 | **36.43 ± 0.48** | 21.12 ± 3.77 | 22.38 ± 0.92 | **35.85 ± 0.45** | 24.30 ± 0.34 | 5.47 |
| Per-answer avg logprob MC32 | — | — | **-0.61 ± 0.01** | -0.61 ± 0.00 | -1.74 ± 0.02 | -0.63 ± 0.00 | -0.78 |
| Per-answer avg logprob | -2.57 ± 0.00 | -2.96 ± 0.09 | **-0.78 ± 0.15** | -0.78 ± 0.10 | -0.97 ± 0.06 | **-0.78 ± 0.11** | -0.78 |
| Perplexity | 13.12 ± 0.00 | 19.40 ± 1.65 | **2.20 ± 0.34** | 2.19 ± 0.22 | 18.86 ± 0.20 | 2.65 ± 0.16 | 2.19 |
| CoT length | 223.27 ± 4.44 | 509.50 ± 40.83 | 448.19 ± 11.23 | 457.67 ± 13.76 | 448.46 ± 9.55 | 405.22 ± 16.92 | 5.00 |

Table 1: **Results on verifiable domains, G=32.** Final performance of models across all our algorithms and metrics. Results are averaged over two seeds. Rows are labeled by the test metrics, columns by the algorithms. We observe that methods which use the log-probability as a reward (Log-prob, Avg Logprop, JEPO) often underperform the baseline when the answer is sampled. However, the gap closes when the answer is produced deterministically (*greedy success*). Perplexity and log-prob based metrics universally improve for the log-prob family of rewards, even surpassing SFT levels, while base RL lags behind in this metric, and probability-based rewards situate themselves in the middle between those. Learning curves are shown in Figures 1 and 5 to 7.

1 and giving the correct answer with probability 0.01 or 0, while this makes a large difference for log-probabilities. Namely, logprob-trained models make sure that if they are wrong, they are not *confidently* wrong, by attributing some nonzero probability to all plausible answers.

Overall, logprob-trained models get both good success rates and good perplexity, while models trained directly for the success rate sacrifice perplexity. Presumably, logprob-trained models smooth out their predictions, while verifier or probability-based variants emit "sharper" probabilities.

### 3.3 RESULTS ON NONVERIFIABLE DOMAINS

We present the results on NuminaProof and Alpaca with the Llama and Qwen models in Table 2 and Figure 2 and Figures 12 to 14 in Appendix C. We observe that training with logprobs, or with average logprobs or JEPO, consistently matches the performance of SFT. As predicted, *Probability* (VeriFree) fails to improve on these metrics, and *Average Probability* (RLPR) is noisier but trails the logprob family closely. This establishes the log-prob family of rewards as a universal method for both verifiable and non-verifiable domains.

### 3.4 LENGTH OF THE CHAIN-OF-THOUGH DURING TRAINING

We now report some intriguing observations on the behavior of the CoT during training, for which we have no complete explanation. In Figure 1, we see that CoTs trained with Logprob variants show an initial dip in length, followed by a recovery in verifiable domains. This pattern does not occur for Prob variants or Base RL. For non-verifiable rewards, we see an even starker pattern in Figure 2: the CoT dips to a length of 10 tokens (including formatting tokens) and never recovers. This means that the CoT is largely eliminated, and Logprob methods effectively become SFT – indeed we observe that the perplexity of methods with a collapsed CoT closely match those of the SFT baseline.

We tried two types of interventions on this pattern: increasing the KL divergence regularization to the base model, and introducing a length penalty that adds a negative reward for every token below

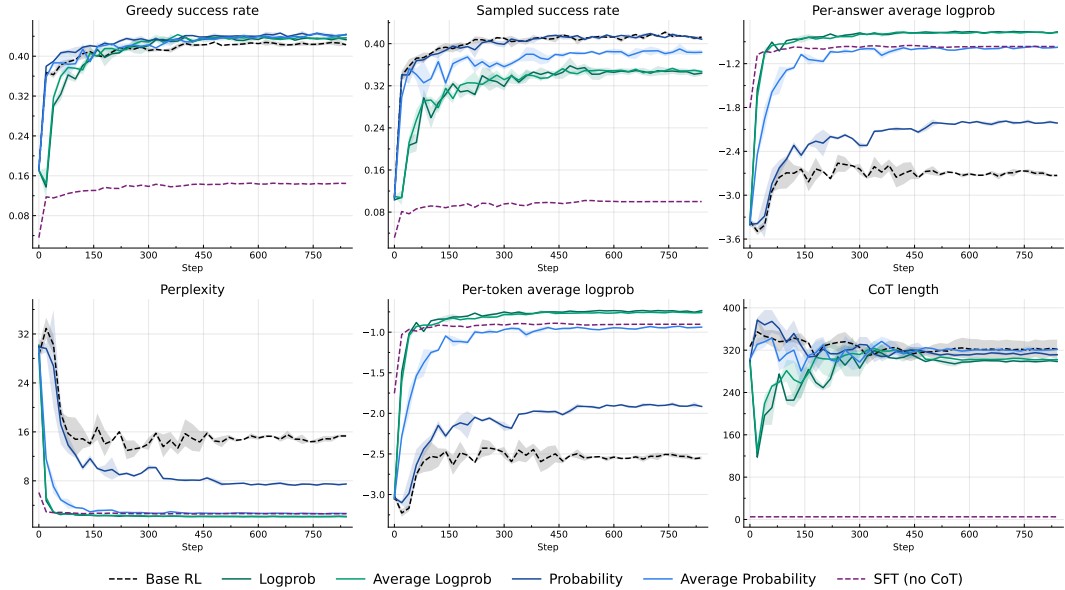

Figure 1: **Verifiable. Llama 3.2 3B Instruct on MATH, G=32.** Learning curves of our algorithms for various metrics. Dashed curves represent the RL baseline and (no-CoT) SFT; green shades for the logprob family of rewards (Logprob, Average logprob and JEPO) and blue for probability-based rewards *Probability (VeriFree)* and *Average Probability* (RLPR). Numerical values can be found in Table 1.

|  | Base model | Log-prob | Avg Logprob | Probability | Avg Probability | JEPO | SFT (no CoT) |
|---|---|---|---|---|---|---|---|
| **Llama 3B, NuminaProof** | | | | | | | |
| Per-answer avg logprob | -1.2871 | -1.1124 | -1.1175 | -1.2859 | -1.1157 | **-1.1104** | -1.1127 |
| Perplexity | 3.62 | **3.04** | 3.06 | 3.62 | 3.05 | **3.04** | **3.04** |
| Per-answer avg logprob MC32 | — | -1.1124 | -1.1175 | -1.2850 | -1.1157 | **-1.1102** | -1.1127 |
| CoT length | 474.0 | 9.1 | 9.0 | 469.6 | 9.1 | 34.1 | 5.0 |
| **Qwen 3B, NuminaProof** | | | | | | | |
| Per-answer avg logprob | -1.1838 | -1.0174 | -1.0235 | -1.1862 | -1.0217 | -1.0218 | **-1.0172** |
| Perplexity | 3.27 | **2.77** | 2.78 | 3.27 | 2.78 | 2.78 | **2.77** |
| Per-answer avg logprob MC32 | — | -1.0174 | -1.0235 | -1.1852 | -1.0217 | -1.0218 | 1.0172 |
| CoT length | 225.6 | 5.3 | 9.0 | 225.2 | 10.0 | 14.1 | 5.0 |
| **Llama 3B, Alpaca** | | | | | | | |
| Per-answer avg logprob | -1.3493 | -0.9397 | -0.9449 | -1.5772 | -0.9513 | -0.9443 | **-0.9381** |
| Perplexity | 3.85 | **2.56** | 2.57 | 4.84 | 2.59 | 2.57 | **2.56** |
| Per-answer avg logprob MC32 | — | -0.9396 | -0.9449 | -1.5724 | -0.9512 | -0.9436 | **-0.9381** |
| CoT length | 134.2 | 14.2 | 14.1 | 58.6 | 9.1 | 14.8 | 5.0 |
| **Qwen 3B, Alpaca** | | | | | | | |
| Per-answer avg logprob | -1.3968 | **-0.8903** | -0.8933 | -1.2982 | -0.8989 | -0.8976 | -0.8905 |
| Perplexity | 4.04 | **2.44** | **2.44** | 3.66 | 2.46 | 2.45 | **2.44** |
| Per-answer avg logprob MC32 | — | **-0.8902** | -0.8931 | -1.2955 | -0.8988 | -0.8969 | -0.8905 |
| CoT length | 83.7 | 16.7 | 14.3 | 16.6 | 15.2 | 16.6 | 5.0 |

Table 2: **Results on non-verifiable domains.** Final performance across all initial models and metrics, on nonverifiable datasets. Probability rewards fail to learn due to their extremely low rewards. We observe that methods which use the log-probability experience a CoT collapse, reducing to SFT. The corresponding learning curves are shown in Figure 2 and Figures 12 to 14 in Appendix C.

a certain threshold in the CoT (see Appendix D for details). These interventions worked in that they prevented the CoT length dip, but this came at the cost of actual performance, as shown in Figures 16 to 19 in Appendix D. It looks like these methods perform best on nonverifiable domains only by reducing the CoT and mimicking SFT. We tried to understand the mechanism behind this dip: we assumed that early in training, shorter CoTs might lead to better predictions since the base model has been trained without CoTs. An initial negative correlation between CoT length and reward may push the model towards shorter CoTs during reinforcement learning. Figures 3 and 4 plot the

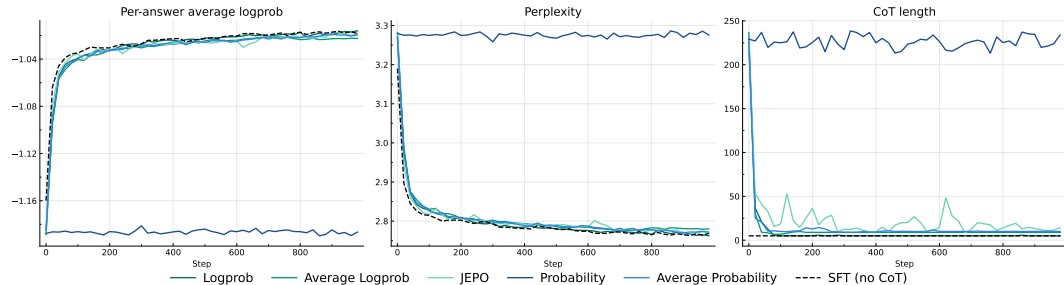

Figure 2: **Non-verifiable: Qwen 2.5 3B Instruct on NuminaProof.** Learning curves of our algorithms for three metrics. Numerical values can be found in Table 2. Log-prob family models match the (per-answer) average log-prob and perplexity from SFT, while probability rewards fail to improve on these metrics due to the sparsity of the rewards. We observe a rapid "collapse" in CoT-length for the log-prob family.

CoT length against the log-probability of the answer for the base model: while the correlation is consistently negligible with nonverifiable answers, it is clearly negative in a verifiable context.

We also considered that the SFT term overpowers the RL part, as the model is not used to writing proofs. To address this, we attempted initializing from a model SFT-ed with masked CoTs to produce a warm start model which can use the CoT effectively. We describe these results in Appendix D, however they also fail to improve on the SFT baseline under a reasonable compute budget.

It is worth noting that in a similar setting, Tang et al. (2025) show JEPO eventually exceeding the SFT performance with long-form answers. The critical difference between our settings is that Tang et al. (2025) train for significantly longer (an order of magnitude) at a larger batch size and lower learning rate. So it is possible that JEPO enables training with long-form answers, at the cost of much higher compute requirements compared to RL with short-form, verifiable answers.

**Discussion: Why does RL eliminate CoTs for nonverifiable domains?** We can put forward several hypothetical explanations for the elimination of CoTs in nonverifiable domains. One possibility is that it takes longer to train long CoTs than short CoTs. RL has a worse signal-to-noise ratio when the number of actions increases, because of the well-known "credit-assignment problem". For CoT training, the actions are the tokens, so it is harder to identify good correlations in long CoTs than short ones. If this is the case, then a correlation between short CoTs and better performance would appear in the early phases of training, only to disappear once long CoTs catch up in performance. This could explain the dip-and-recovery pattern for verifiable domains, even if initially no such correlation exists. However, this does not explain why this pattern occurs for Log-prob but not for Prob in verifiable domains. Also, in this situation, we would expect the length penalties to help.

Another possibility is that, with long answers, the model has time to deploy a hidden CoT within its internal layers. Indeed, the survey by Zhu et al. (2025) puts forward increasing evidence for the existence of such hidden CoTs in LLMs. Conversely, for the very short answers in verifiable domains, there may be too few tokens to have an efficient internal CoT during the answer, and an actual, non-hidden CoT may be necessary. If this is the case, it may be interesting to build datasets that interpolate between short and long answers, and see if there is a transition. We leave these investigations to future work.

## 4 CONCLUSION AND DISCUSSION

Our work establishes log-probability rewards as a unifying training signal effective in both verifiable and non-verifiable domains, without relying on ground-truth correctness labels. On reasoning benchmarks like MATH and DeepScaleR, log-probability rewards match the success rates of standard 0/1 RL objectives while substantially improving perplexity; on long-form proofs, they match supervised fine-tuning while other probability-based variants fall well below. This shows that the same criterion can be carried seamlessly across settings. This highlights their potential as a general recipe for post-training reasoning LLMs, valid over the full range of possible answer types. In future work, we hope to hope to further develop this approach on non-verifiable domains, enabling efficient RL training on any dataset, leveraging the CoT capabilities.

**Reproducibility Statement**  To enable independent re-implementation, we provide in the paper and Appendix: (i) datasets and base models used for training and evaluation (ii) full hyperparameter details and training objective; (iii) prompt template and inference settings for evaluation; (iv) verifier used; (v) number of runs per result, with mean ± std; and (vi) compute details.

**LLM Usage**  We used LLMs for drafting and polishing the writing. The research itself, and the bulk of the writing effort, were done by humans.

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

## A  LOSSES AND ADVANTAGES FOR THE REWARDS CONSIDERED

**Lemma 1.** *Let $z$ be a chain-of-thought variable sampled from a model $\pi_\theta$ with parameters $\theta$, and let $R_\theta(z)$ be a reward function that depends on $z$ and also possibly on $\pi_\theta$ (for instance, $R_\theta(z) = \log \pi_\theta(a^\star|z)$ or $R_\theta(z) = \pi_\theta(a^\star|z)$).*

*Then the expected reward*

$$J_\theta = \mathbb{E}_{z \sim \pi_\theta}[R_\theta(z)] \tag{19}$$

*has the same gradients (up to sign) as the loss function*

$$L_\theta = \mathbb{E}_{z \sim \pi_\theta^{\text{sg}}} \left[ (R_\theta(z) - c_\theta)^{\text{sg}} \log \pi_\theta(z) + R_\theta(z) \right] \tag{20}$$

*where* $^{\text{sg}}$ *denotes a stop-grad operator, and* $c_\theta$ *is any expression independent of* $z$.

For instance, in RLOO, $c_\theta$ is the average of $R_\theta(z')$ over samples $z' \sim \pi_\theta$ independent from $z$.

*Proof.* The gradient of $J_\theta$ is

$$\nabla_\theta \mathbb{E}_{z \sim \pi_\theta} [R_\theta(z)] = \nabla_\theta \sum_z \pi_\theta(z) R_\theta(z) \tag{21}$$

$$= \sum_z (\pi_\theta(z) \nabla_\theta \log \pi_\theta(z)) R_\theta(z) + \sum_z \pi_\theta(z) \nabla_\theta R_\theta(z) \tag{22}$$

$$= \mathbb{E}_{z \sim \pi_\theta} [R_\theta(z) \nabla_\theta \log \pi_\theta(z)) + \nabla_\theta R_\theta(z)] \tag{23}$$

hence the statement without $c_\theta$.

Now we have $\mathbb{E}_{z \sim p_\theta} \nabla_\theta \log \pi_\theta(z) = \sum_z \pi_\theta(z) \nabla_\theta \log \pi_\theta(z) = \sum_z \nabla_\theta \pi_\theta(z) = \nabla_\theta 1 = 0$. Therefore, we have $\mathbb{E}_{z \sim \pi_\theta} [c_\theta \nabla_\theta \log \pi_\theta(z)] = 0$ as long as $c_\theta$ is independent of $z$. Hence we can subtract $c_\theta \nabla_\theta \log \pi_\theta(z)$ from the expression above, which leads to the conclusion. $\square$

# B  EXPERIMENTAL DETAILS

For each experiment, we use a synchronous implementation of RLOO running in parallel across 8 processes. We use the AdamW (Kingma & Ba, 2014) optimizer with a learning rate of $10^{-5}$, and a cosine schedule with a 20 step warm-up. During our research, we tried a few learning rates for the probability rewards, but noticed that the chosen value worked consistently for all variants. We clip the global gradient norm to a global threshold of $1.0$. Each batch contains 8 questions from the dataset with $G$ different CoTs; such a batch corresponds to one *step* in all our figures.

**Full Details on the Datasets.** We consider two *verifiable* math benchmarks and two *non-verifiable* long-form datasets. (i) **MATH** (Hendrycks et al., 2021b): we concatenate all official subsets, parse the final answer from \boxed{...}, discard intermediate solutions, and hold out a random 10% for validation. We report accuracy on the official test split. The resulting training set contains ~7,000 short-answer problems. (ii) **DeepScaleR (Preview)** (Luo et al., 2025): we discard long solutions, use the provided final answer as ground truth, hold out a random 10% for validation, and report performance on this held-out set. The training set has ~39,000 short-answer problems. (iii) **Alpaca (cleaned)** (Taori et al., 2023): we use the standard cleaned variant; 1,000 random examples are used for validation, leaving ~50,000 training samples with predominantly long-form answers. (iv) **NuminaProof**: starting from NUMINAMATH-1.5 (Li et al., 2024), we filter for theorem–proof style items (full solutions are proofs), remove instances with hyperlinks, and sanitize the remaining solutions. We reserve 1,000 examples for validation, yielding ~50,000 long-form training samples.

**Prompting and formatting.** All experiments use a DEEPSEEK-R1–style instruction format (Guo et al., 2025) with the instruction as the system prompt and the question as the user message, rendered with each model family's standard instruct template (Llama or Qwen). We prefill the assistant turn with "<think>" to initiate the reasoning trace. The final sentence of the system prompt—encouraging concise, easily parsable answers—is enabled only in verifiable settings (see Template 1).

At each training step, each process receives a question prompt, and generates $G$ completions with a maximum length of $T$ tokens to that question. Unless noted otherwise, we use $G = 32$ in verifiable domains, and $G = 4$ in nonverifiable domains, and $T = 1024$. We generate completions until they reach the pattern </answer, but for the likelihood-based rewards, we truncate the CoT at <answer. This is inspired by Zhou et al. (2025) who pointed out that in both the Llama and Qwen tokenizers, there is no individual token that contains the pattern r>, and thus it is guaranteed to be a consistent token boundary.

For Base RL, the verifier tries to parse <answer>answer</answer> and match it with the ground truth. If the answer is correct, the reward is 100. If the answer is incorrect but the format

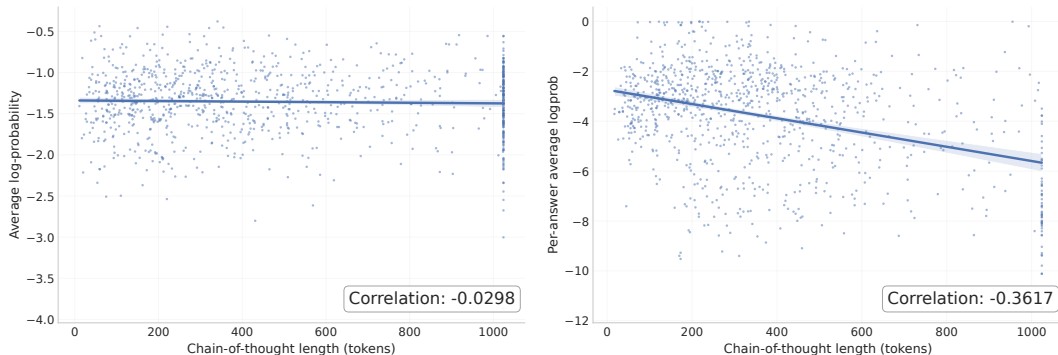

Figure 3: A scatterplot displaying the relation between CoT length and the probability of obtaining the correct answer, when generating the CoT with **Llama 3.2 3B Instruct**. **Left: NuminaProof.** We observe a low correlation, leaning negative. **Right: DeepScaleR.** We observe a clear negative correlation. This explains the initial dip in the CoT length on verifiable domains, but does not explain the collapse on nonverifiable tasks.

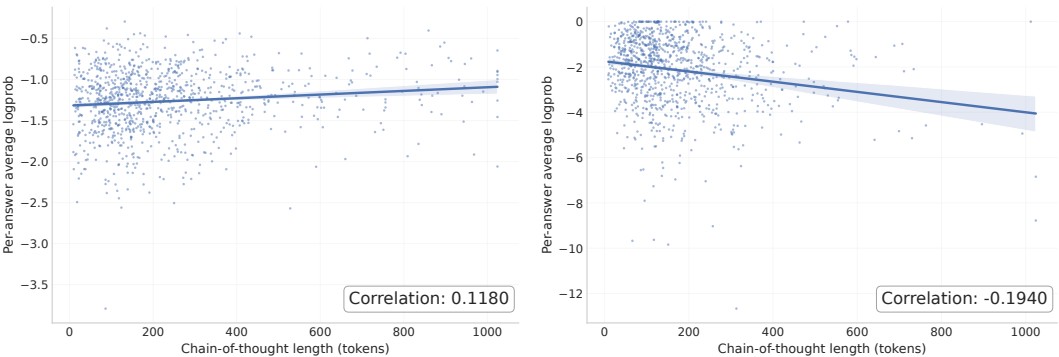

Figure 4: A scatterplot displaying the relation between CoT length and the probability of obtaining the correct answer, when generating the CoT with **Qwen 2.5 3B Instruct**. **Left: NuminaProof.** We observe a low correlation, leaning positive. **Right: DeepScaleR.** We observe a clear negative correlation.

is kept correctly (parsing was succesful), the reward is 10. If the format is incorrect and an answer cannot be parsed, the reward is 0. We train and evaluate with *exact* match on the answer.

> **Template 1** (System prompt). *A conversation between User and Assistant. The user asks a question, and the Assistant solves it. The assistant first thinks about the reasoning process in the mind and then provides the user with the answer. The reasoning process and answer are enclosed within <think></think> and <answer></answer> tags, respectively, i.e., <think>reasoning process here</think> <answer>answer here</answer>. Inside the answer tag, put only the answer and no additional commentary.*

## C  ADDITIONAL EXPERIMENTAL RESULTS

### C.1  LENGTH-LOGPROB CORRELATIONS

Here we provide scatterplots that visualize the correlations between CoT length and correct answer logprob, as discussed in Section 3.4.

## C.2 VERIFIABLE DOMAINS

Here, we complement Figure 1 with the corresponding learning curves for other model-dataset combinations (Figures 5 to 7) and provide the corresponding Figures 8 to 11 and Table 3 for training with $G = 4$ (including JEPO, which for efficiency reasons we only ran for $G = 4$).

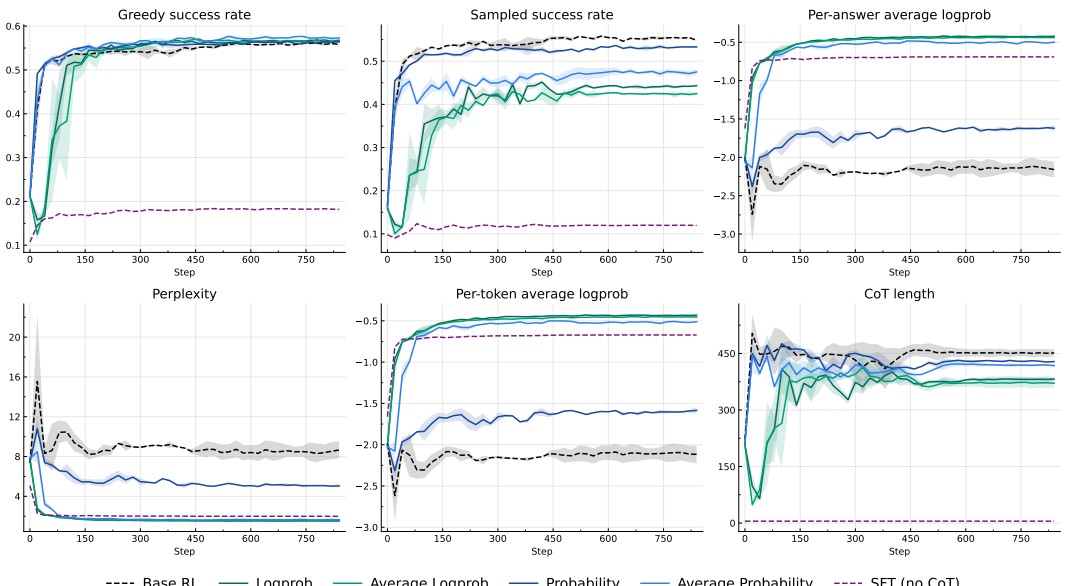

Figure 5: **Verifiable. Qwen 2.5 3B Instruct on MATH with a group size of 32.**

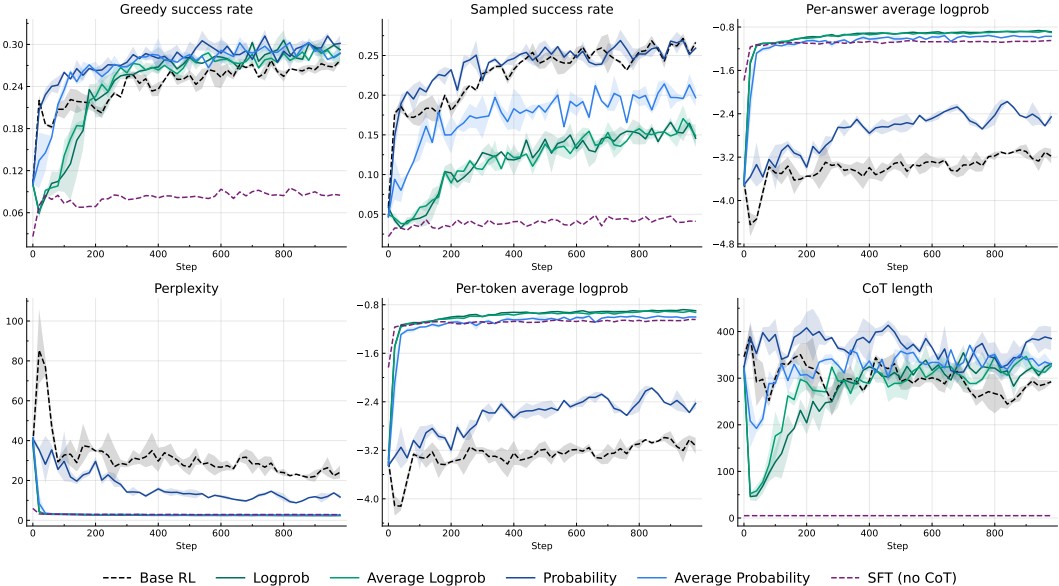

Figure 6: **Verifiable. Llama 3.2 3B Instruct on DeepScaleR with a group size of 32.**

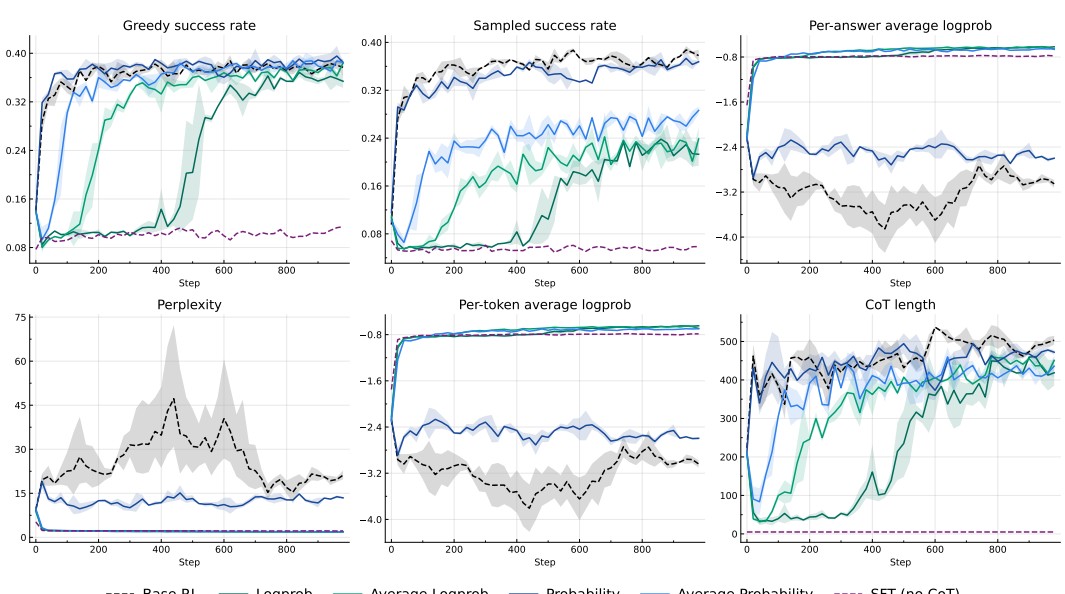

Figure 7: **Verifiable. Qwen 2.5 3B Instruct on DeepScaleR with a group size of 32.**

|  | Base model | Base RL | Log-prob | Avg Logprob | JEPO | Probability | Avg Probability | SFT (no CoT) |
|---|---|---|---|---|---|---|---|---|
| **Llama 3B, MATH** | | | | | | | | |
| Greedy success | 17.13 ± 0.00 | **42.09 ± 1.08** | 40.58 ± 2.63 | 40.52 ± 0.01 | **43.19 ± 0.10** | 40.48 ± 0.42 | 41.56 ± 0.52 | 14.38 |
| T=1 Sampled Success | 10.77 ± 0.07 | **40.34 ± 1.30** | 31.81 ± 1.40 | 31.06 ± 1.82 | 35.85 ± 0.80 | 38.03 ± 0.99 | 35.03 ± 0.73 | 9.96 |
| Per-answer avg logprob MC32 | — | — | **-0.72 ± 0.02** | **-0.73 ± 0.02** | -0.79 ± 0.01 | -1.52 ± 0.04 | -0.82 ± 0.00 | -0.96 |
| Per-answer avg logprob | -4.21 ± 0.00 | -2.98 ± 0.06 | **-0.83 ± 0.01** | -1.02 ± 0.01 | -1.23 ± 0.04 | -2.56 ± 0.06 | -1.11 ± 0.01 | -0.96 |
| Perplexity | 67.29 ± 0.00 | 19.65 ± 1.17 | **2.30 ± 0.03** | 2.77 ± 0.04 | 3.41 ± 0.14 | 12.97 ± 0.81 | 3.03 ± 0.04 | 2.62 |
| CoT length | 331.66 ± 2.02 | 373.52 ± 23.15 | 283.84 ± 27.47 | 305.76 ± 40.62 | 326.15 ± 21.00 | 345.83 ± 10.13 | 333.47 ± 45.09 | 5.00 |
| **Qwen 3B, MATH** | | | | | | | | |
| Greedy success | 21.19 ± 0.00 | 53.19 ± 0.18 | 51.70 ± 2.65 | 53.87 ± 0.24 | 54.01 ± 2.10 | **54.59 ± 0.24** | 53.52 ± 1.47 | 18.32 |
| T=1 Sampled Success | 16.10 ± 0.06 | **50.31 ± 0.03** | 36.36 ± 1.71 | 37.53 ± 1.32 | 42.55 ± 3.69 | 48.75 ± 0.77 | 42.37 ± 2.22 | 12.00 |
| Per-answer avg logprob MC32 | — | — | **-0.47 ± 0.00** | -0.46 ± 0.01 | -0.47 ± 0.02 | -1.00 ± 0.03 | -0.49 ± 0.02 | -0.69 |
| Per-answer avg logprob | -2.19 ± 0.00 | -1.89 ± 0.01 | **-0.67 ± 0.09** | -0.68 ± 0.01 | -0.78 ± 0.02 | -1.77 ± 0.02 | -0.79 ± 0.11 | **-0.69** |
| Perplexity | 8.92 ± 0.00 | 6.60 ± 0.03 | **1.97 ± 0.19** | 1.98 ± 0.02 | 2.18 ± 0.05 | 5.85 ± 0.13 | **2.22 ± 0.24** | 1.99 |
| CoT length | 222.43 ± 0.80 | 429.31 ± 0.17 | 326.69 ± 49.32 | 387.34 ± 7.17 | 362.67 ± 70.73 | 447.37 ± 34.30 | 380.90 ± 10.57 | 5.00 |
| **Llama 3B, DeepScaleR** | | | | | | | | |
| Greedy success | 10.05 ± 0.00 | 23.95 ± 0.92 | **25.28 ± 5.14** | 24.51 ± 0.93 | **26.35 ± 1.13** | 25.78 ± 0.11 | 25.14 ± 1.22 | 9.68 |
| T=1 Sampled Success | 4.80 ± 0.42 | **18.40 ± 1.38** | 12.34 ± 3.55 | 12.74 ± 0.45 | 16.34 ± 2.27 | 21.91 ± 2.28 | 15.92 ± 0.06 | 4.53 |
| Per-answer avg logprob MC32 | — | — | **-0.90 ± 0.03** | -0.91 ± 0.01 | -0.91 ± 0.01 | -1.67 ± 0.21 | -0.95 ± 0.02 | -1.07 |
| Per-answer avg logprob | -4.92 ± 0.00 | -3.02 ± 0.03 | **-0.97 ± 0.03** | -1.21 ± 0.19 | -1.15 ± 0.10 | -2.77 ± 0.41 | -2.26 ± 0.17 | -1.07 |
| Perplexity | 137.25 ± 0.00 | 20.43 ± 0.69 | **2.64 ± 0.08** | 3.38 ± 0.65 | 3.16 ± 0.31 | 16.66 ± 6.64 | 9.67 ± 1.60 | 2.91 |
| CoT length | 348.50 ± 2.22 | 315.51 ± 60.02 | 328.66 ± 127.51 | 282.46 ± 40.37 | 351.95 ± 82.12 | 323.45 ± 32.93 | 326.47 ± 23.48 | 5.00 |
| **Qwen 3B, DeepScaleR** | | | | | | | | |
| Greedy success | 14.02 ± 0.00 | **37.00 ± 0.57** | 28.46 ± 0.46 | 31.42 ± 3.67 | 30.14 ± 9.20 | 35.17 ± 2.16 | 33.68 ± 0.45 | 11.33 |
| T=1 Sampled Success | 9.15 ± 0.07 | **33.87 ± 0.02** | 15.33 ± 0.46 | 16.91 ± 3.19 | 20.17 ± 5.69 | 28.76 ± 2.77 | 20.90 ± 0.98 | 5.62 |
| Per-answer avg logprob MC32 | — | — | **-0.67 ± 0.00** | -0.67 ± 0.04 | -0.68 ± 0.04 | -1.44 ± 0.03 | **-0.67 ± 0.00** | -0.77 |
| Per-answer avg logprob | -2.57 ± 0.00 | -2.83 ± 0.24 | -1.12 ± 0.28 | -1.23 ± 0.47 | **-0.99 ± 0.02** | -2.62 ± 0.08 | -1.23 ± 0.19 | **-0.77** |
| Perplexity | 13.12 ± 0.00 | 17.12 ± 4.02 | 3.15 ± 0.95 | 3.61 ± 1.65 | **2.68 ± 0.05** | 13.82 ± 1.13 | 3.45 ± 0.67 | 2.17 |
| CoT length | 222.45 ± 1.24 | 506.71 ± 57.31 | 336.45 ± 111.93 | 368.17 ± 37.71 | 349.85 ± 89.50 | 392.60 ± 76.07 | 390.58 ± 59.18 | 5.00 |

Table 3: **Results on verifiable domains, G=4.** Final performance of models trained with a group size of 4, across all our algorithms (including JEPO) and metrics. Conclusions mirror those of Table 1. The corresponding learning curves are presented in Figures 8 to 11.

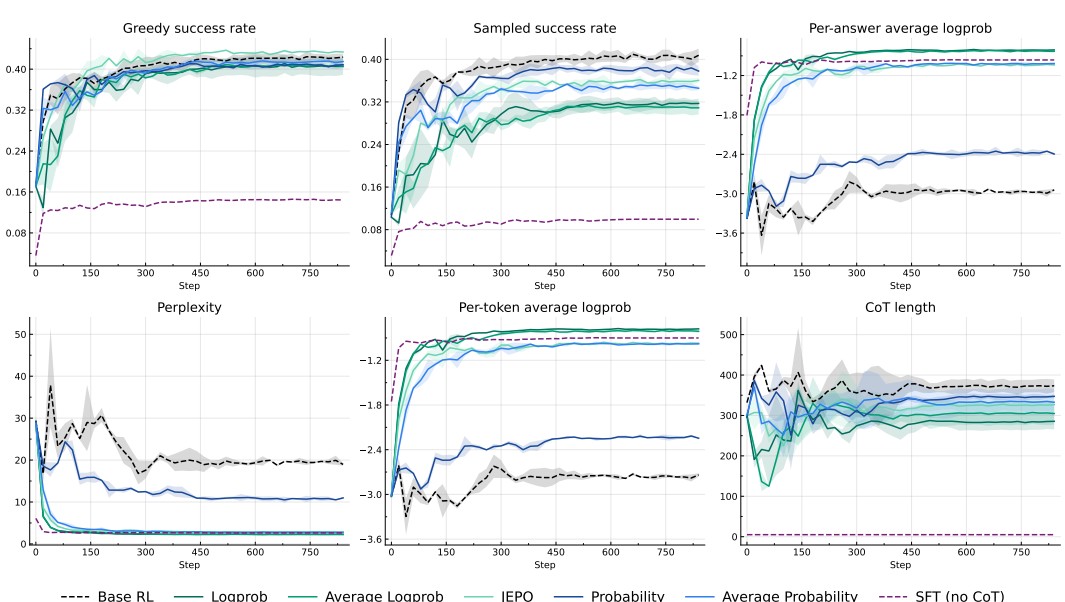

Figure 8: **Verifiable: Llama-3.2-3B on MATH with a group size of 4.**

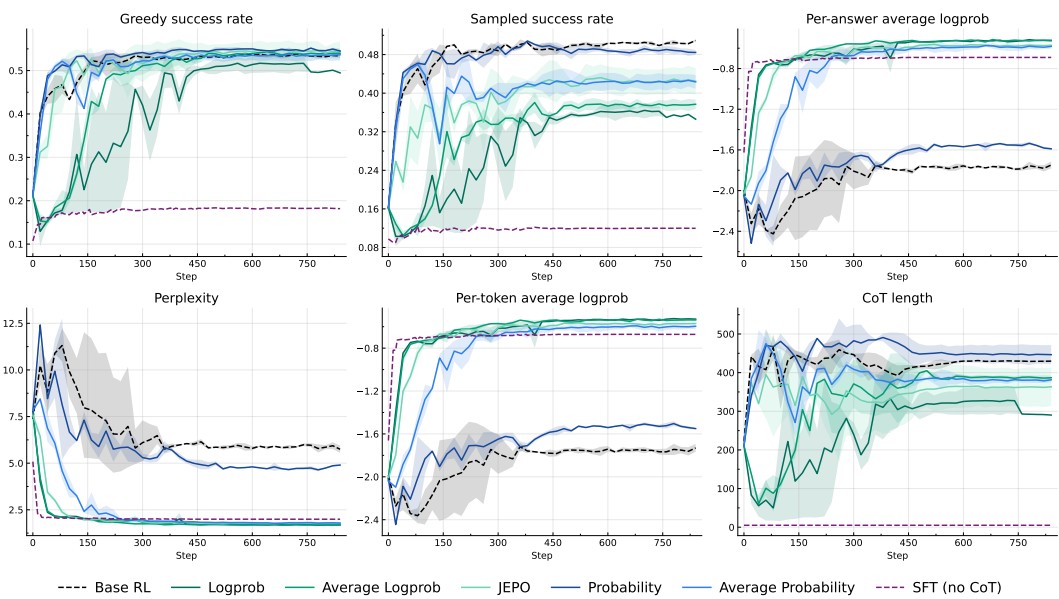

Figure 9: **Verifiable. Qwen 2.5 3B Instruct on MATH with a group size of 4.**

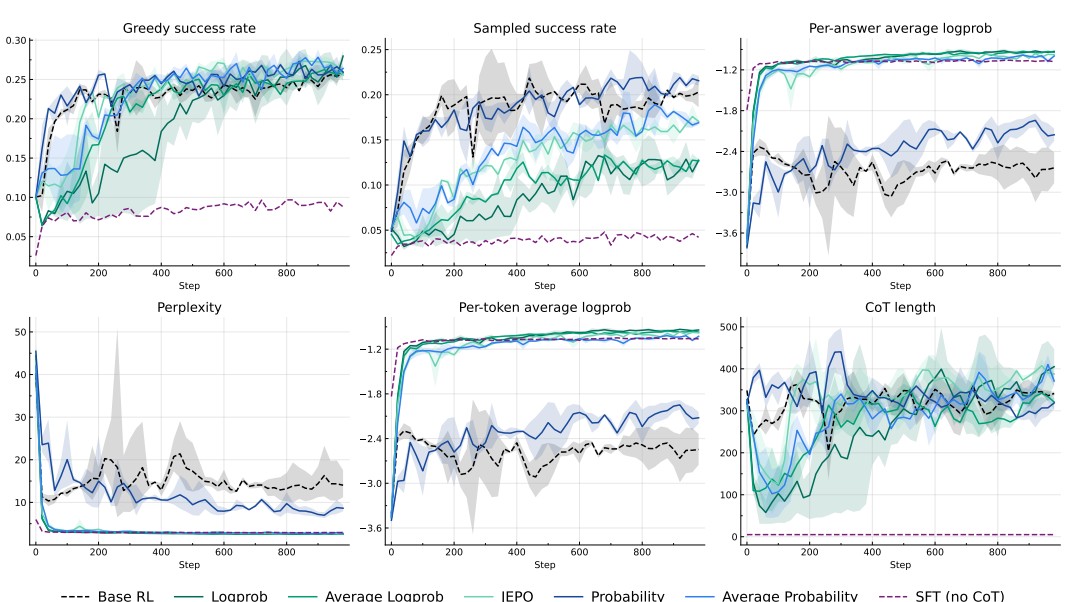

Figure 10: **Verifiable. Llama 3.2 3B Instruct on DeepScaleR with a group size of 4.**

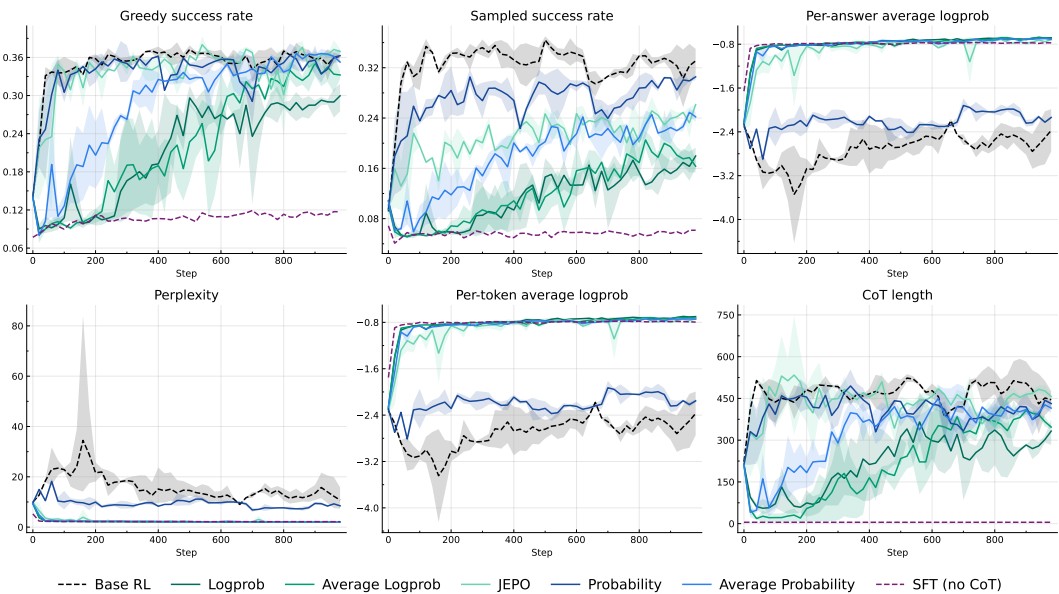

Figure 11: **Verifiable. Qwen 2.5 3B Instruct on DeepScaleR with a group size of 4.**

## C.3 NON-VERIFIABLE DOMAINS

Here we provide the Figures complementary to Figure 2 for other model/dataset combinations in Figures 12 to 14.

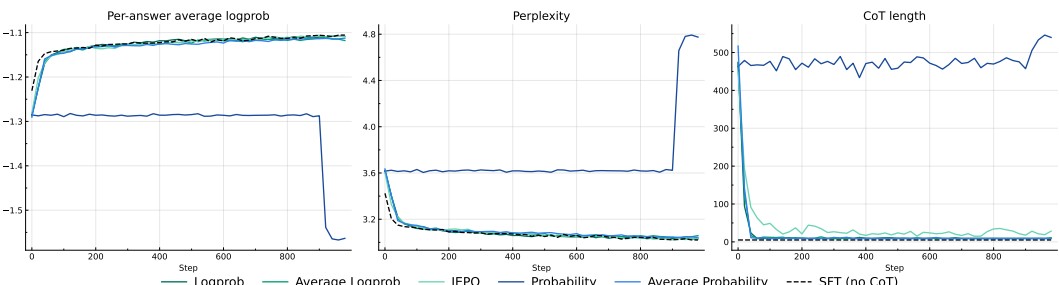

Figure 12: **Non-verifiable. Llama 3.2 3B Instruct on NuminaProof.**

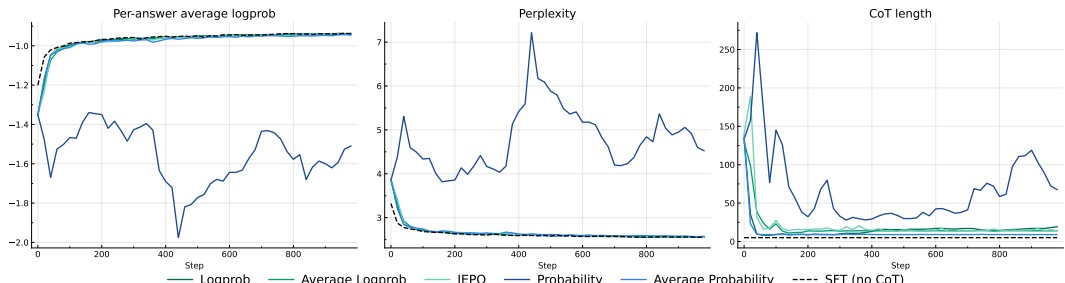

Figure 13: **Non-verifiable. Llama 3.2 3B Instruct on Alpaca.**

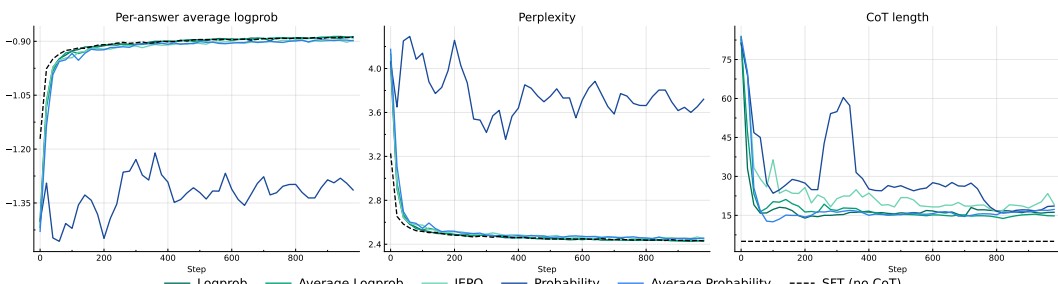

Figure 14: **Non-verifiable. Qwen 2.5 3B Instruct on Alpaca.**

# D ATTEMPTED REGULARIZATION METHODS

In Section 3.4 we use two types of regularization to stabilize the CoT in nonverifiable domains. The first is straight-forward – we include a KL divergence term in the loss, which keeps the model close to the initial model, as proposed by Guo et al. (2025).

The second type introduces an additional reward term:

$$R_l(z) = r \cdot \min\{|z| - l_0, 0\}$$

that is, for each missing token below a threshold for $l_0$, a negative reward $r$ is applied. We vary the threshold $l_0$ and report results for values of $100, 150, 300$ and $500$. To set the value of $r$, we design it so that it approximately compensates the increase of the reward during the initial CoT length drop over the initial 40 training steps. Specifically, we take the base nonverifiable experiments for each algorithm, model and dataset, and set

$$r = \frac{\Delta R}{\Delta L}$$

where $\Delta R$ is the increase in validation reward, and $\Delta L$ is the decrease in the average CoT length.

The results of these ablations are presented in Figures 16 to 19.

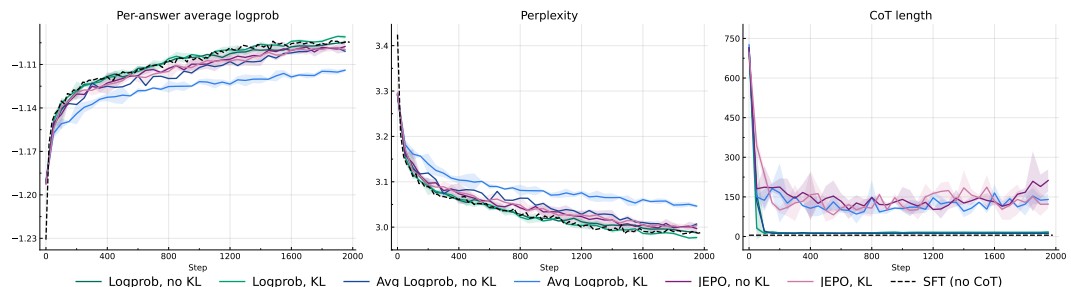

Figure 15: **Non-verifiable. Llama 3.2 3B Instruct-Warmstart on Numina.**

**Warm start**   When training the warm start models, our goal was to improve the initial performance of the model. We observed that at initialization the perplexity of the correct answer is better if it is appended directly after the question, and worse if there is an autoregressively generated CoT between them. To this end, we generated a static dataset of CoTs generated from the initial model, and trained it with SFT on (question, completion, answer) triples, masking out everything but the answer with the intuition that this would train the model to use CoTs.

We display the results in Figure 15. Most variants still collapse the CoT and achieve a performance comparable with SFT. The variants that are regularized to not collapse have a performance lower than SFT. One interesting case is unregularized JEPO, which maintains a CoT of about 100 tokens, and almost matches the performance of SFT.

It is however important to keep in mind that due to the initial warm start phase, this comparison is no longer "fair" in terms of the data used in training, as the SFT baseline did not undergo the warm start mid-training.

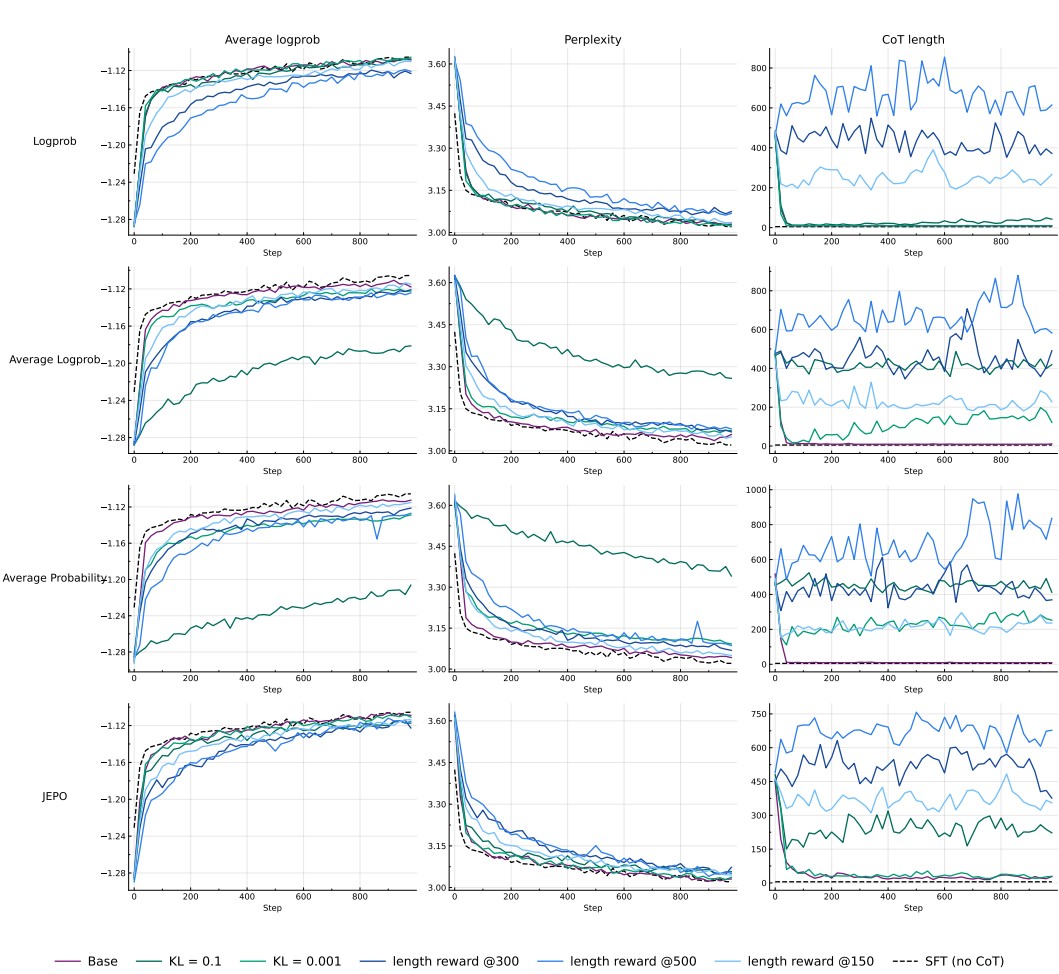

Figure 16: **Non-verifiable. Llama 3.2 3B Instruct on NuminaProof.** Training curves of various attempts at stabilizing the CoT on nonverifiable domains with Llama 3.2 3B on NuminaProof. When the KL divergence coefficient, or the length threshold for the penalty are increased, the CoT does better at maintaining a non-trivial length. However, the actual log-prob of the correct answer decreases accordingly.

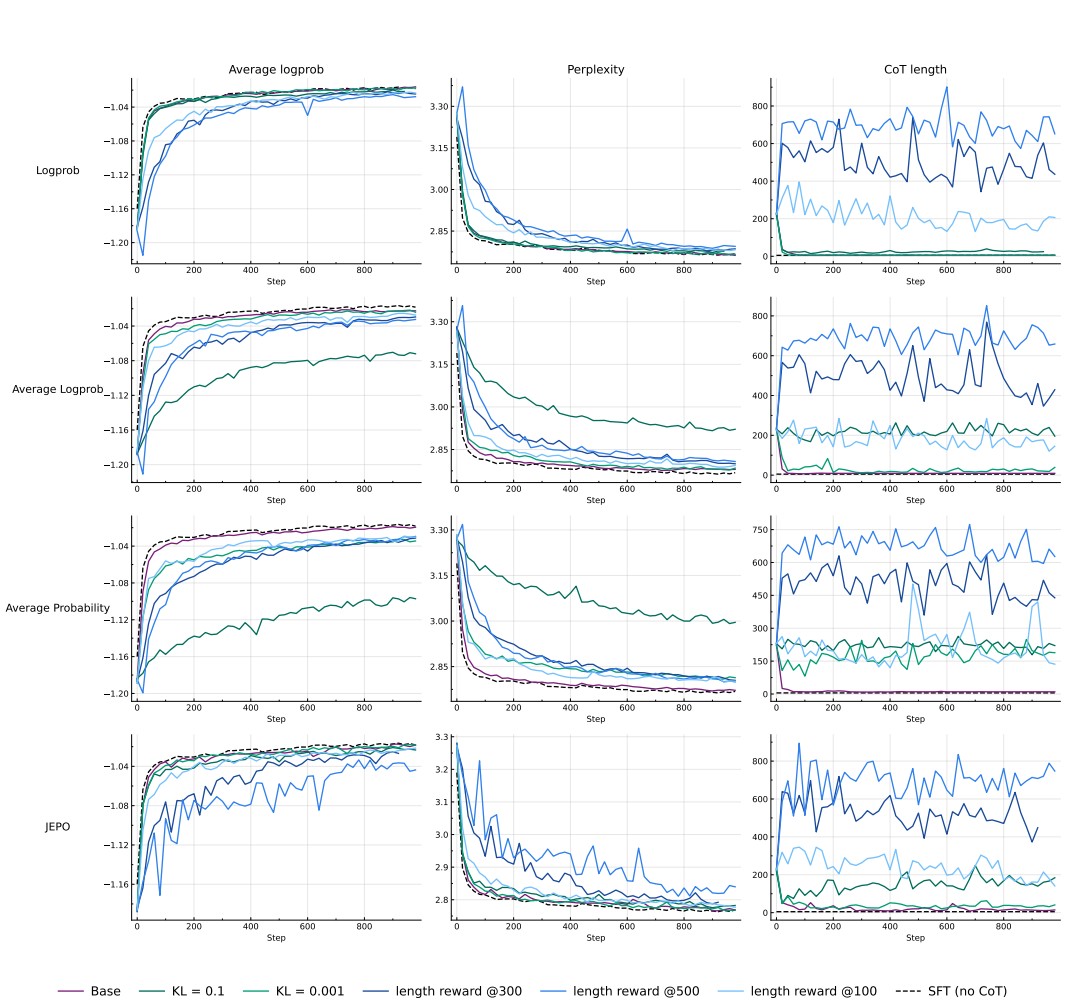

Figure 17: **Non-verifiable. Qwen 2.5 3B Instruct on NuminaProof.** Conclusions are similar to Figure 16.

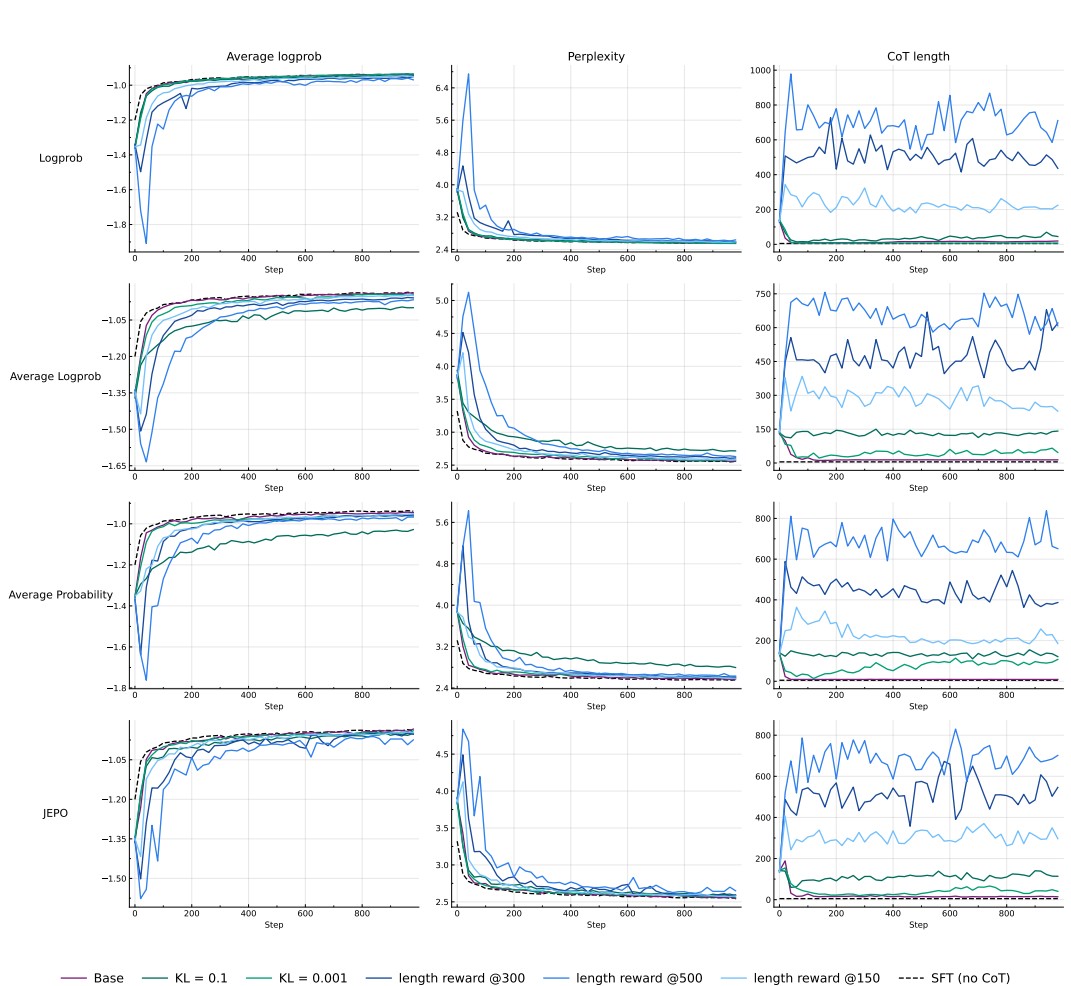

Figure 18: **Non-verifiable. Llama 3.2 3B Instruct on Alpaca.** Conclusions are similar to Figure 16.

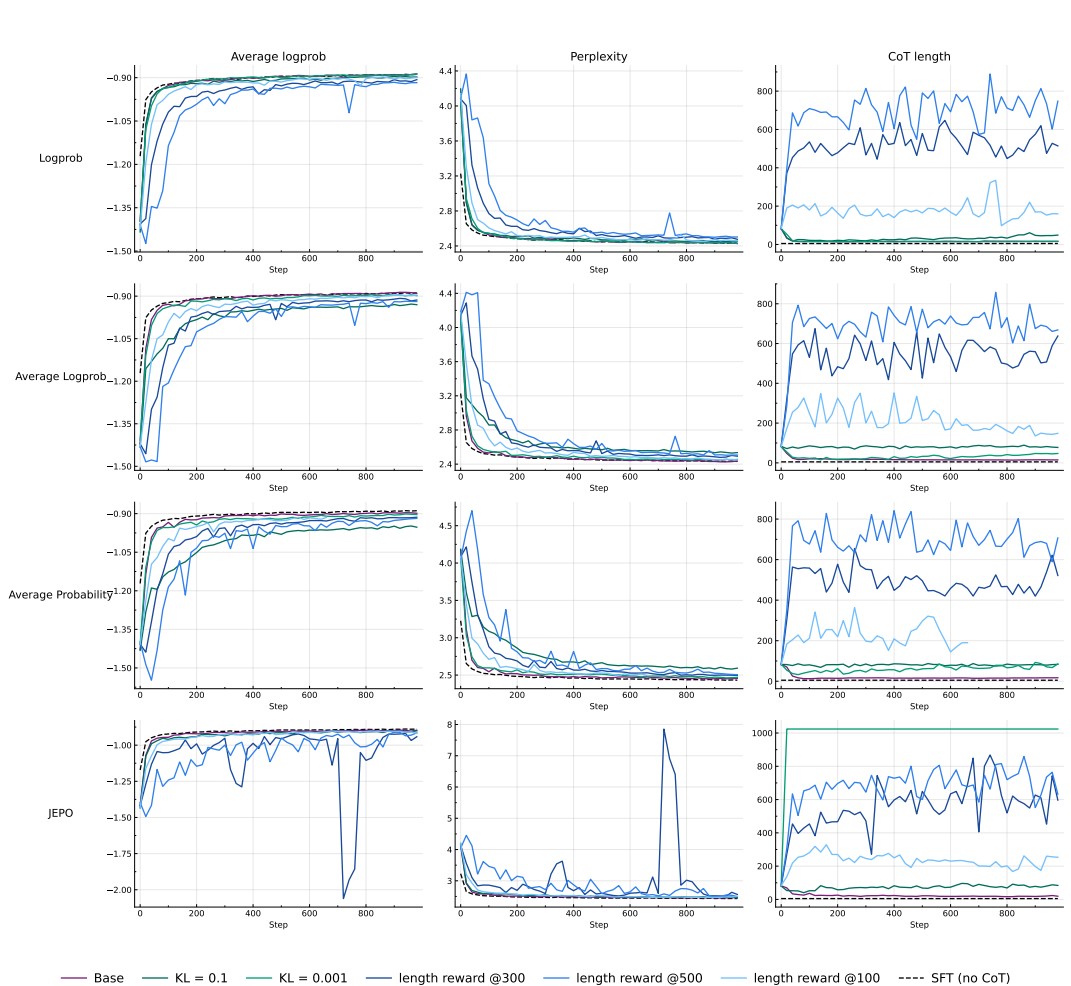

Figure 19: **Non-verifiable. Qwen 2.5 3B Instruct on Alpaca.** Conclusions are similar to Figure 16.

