# OpenReview forum: "Universal Likelihood Rewards for LLM Reasoning"
_ICLR.cc/2026/Conference — ICLR 2026 Conference Withdrawn Submission_

### Official Review · Reviewer_qY12 · 2025-10-17

**Soundness:** 2
**Presentation:** 2
**Contribution:** 2
**Rating:** 4
**Confidence:** 4

**Summary:**

This paper explores **reward design beyond binary 0/1 signals** in LLM reinforcement learning, studying how denser reward schemes can affect training. It provides an empirical comparison of multiple reward formulations and shows that richer reward signals can lead to better performance and faster convergence.

**Strengths:**

It’s definitely worth studying **reward design for LLM RL** beyond binary feedback, since denser rewards can provide stronger learning signals and potentially improve both performance and convergence speed. This paper delivers a solid **empirical study** comparing different reward formulations in this space.

**Weaknesses:**

+ The paper mainly reports empirical findings but provides **limited deeper insights**. It would be more engaging to see either a theoretical explanation or a more detailed empirical breakdown of *why* certain reward designs work better in specific ways.
+ Some **technical details are important** and deserve discussion. In short, some commonly-adopted techniques can mitigate the disadvantage of 0/1 reward, and as people almost always use them, it would be nice to do experiments with them too:
  - Do you use **dynamic filtering for GRPO** (i.e., filtering out prompts with zero-variance rewards)? Without it, the 0/1 reward baseline is at a disadvantage since all-0 or all-1 prompts contribute nothing to the gradient, effectively reducing batch size. This issue is naturally less severe for your dense reward, but since dynamic filtering is now common in LLM RL, it’d be nice to see results with it enabled.
  - Similarly, using **D = 4** for GRPO may be too small. As shown in [https://arxiv.org/abs/2510.01180](https://arxiv.org/abs/2510.01180), increasing D raises the chance that a prompt contributes meaningfully to the gradient. Recent setups often use D ≥ 16, which would make the comparison fairer.

**Questions:**

It would be nice to also **cite [https://arxiv.org/abs/2505.11080](https://arxiv.org/abs/2505.11080)** and discuss how your approach connects or differs from it.

---

### Official Review · Reviewer_hVRZ · 2025-10-31

**Soundness:** 2
**Presentation:** 2
**Contribution:** 2
**Rating:** 2
**Confidence:** 4

**Summary:**

This paper studies probability-based reinforcement learning (RL) rewards for chain-of-thought (CoT) training for both verifiable and non-verifiable tasks. It compares probabilities-based rewards against standard binary rewards and supervised fine-tuning. The main findings include log-probability rewards overall generalize across settings and tasks, by achieving best performance and stable CoT behavior.

**Strengths:**

- The paper presents a comprehensive and systematic study of RL training for chain-of-thought reasoning, examining various reward formulations derived from the (log-)probabilities of reference answers.

- It offers several insightful empirical findings. Notably, using log-probability as the reward signal proves consistently more effective across tasks. In addition, the work highlights a interesting phenomenon where RL training tends to shorten or even eliminate CoTs in non-verifiable domains, while CoTs recover naturally in verifiable settings.

**Weaknesses:**

- The paper is primarily empirical and does not introduce a novel training algorithm or method beyond analyzing different probability-based reward formulations.

- The proposed approaches depend on the availability of reference or ground-truth answers, which restricts their applicability to tasks where such data exist. This dependency limits the diversity and scalability of training data, especially for non-verifiable tasks with multiple valid answers.

- The evaluation can be strengthened by including additional task performance metrics such as accuracy or pass@k, beyond the current set of reported measures.

- The presentation of results needs improvement. Tables 1 and 2 are difficult to interpret, with unclear labeling, multiple boldfaced entries per row and column, and some missing values (e.g., in Table 2). Clearer explanations are needed for the reported metrics and the missing results.

**Questions:**

Please see weaknesses.

---

### Official Review · Reviewer_F7WB · 2025-10-31

**Soundness:** 2
**Presentation:** 1
**Contribution:** 1
**Rating:** 2
**Confidence:** 5

**Summary:**

This manuscript investigates the use of log-probability as a universal reward for RL post-training of LLMs. Specifically, the authors evaluate the effectiveness of log-probability–based rewards across multiple LLMs on both verifiable and non-verifiable tasks. Experimental results demonstrate that log-probability consistently outperforms alternative reward formulations, such as raw probability, highlighting its potential as a more stable and generalizable reward metric for LLM post-training.

**Strengths:**

This manuscript conducts extensive experiments comparing probability and log-probability as reward signals in LLM RL post-training. The results, benchmarked against SFT and Base RL methods, demonstrate the superiority of using log-probability as the reward formulation.

**Weaknesses:**

1. My primary concern lies in the writing and organization of the manuscript. The presentation lacks clarity, and several tables are difficult to interpret. For instance, in Tables 1 and 2, the key metric of accuracy is overshadowed by many less relevant metrics such as perplexity, which distract from the main results. In my opinion, the current version is not yet ready for publication at ICLR.


2. The conclusion is also insufficiently supported. The paper claims that log-probability can serve as a universal reward signal for LLM RL post-training; however, the experimental results show only marginal improvements over the base RL baseline. Moreover, the evaluation is limited to 3B-scale models and a small set of datasets, which weakens the overall persuasiveness of the conclusion.

**Questions:**

Please see Weaknesses.

---

### Official Review · Reviewer_fchY · 2025-11-02

**Soundness:** 3
**Presentation:** 2
**Contribution:** 2
**Rating:** 2
**Confidence:** 3

**Summary:**

This paper systematically evaluates probability-based rewards for reinforcement learning in LLMs, focusing on using the likelihood of the reference answer conditioned on the generated CoT. It compares log-probability, average log-probability, probability (VeriFree), average probability (RLPR), and JEPO against base RL (RLOO) and SFT across verifiable math tasks (MATH, DeepScaleR) and non-verifiable long-form datasets (Alpaca, a proof subset of NuminaMath), using Llama-3.2-3B and Qwen-2.5-3B. Metrics include success rates, log-likelihood/perplexity, and CoT length. The authors finally claimed that log-probability rewards are the only variant that works well across both regimes.

**Strengths:**

1. Clear and well-scoped research topic.

2. Thorough empirical comparison. Multiple reward designs (log-prob, avg-log-prob, prob, avg-prob, JEPO) against both RL and SFT baselines. Multiple metrics, including perplexity/log-prob estimates with MC approximations.

3. Insightful analysis of training dynamics.

**Weaknesses:**

1. Limited novelty in the core idea. Prior works (VeriFree, RLPR, JEPO) have explored likelihood-based signals. The main novelty here is a careful, unified comparison and the empirical conclusion that log-prob rewards are the most robust across regimes. This is valuable but incremental.

2. The evaluation metric of perplexity on non-verifiable tasks is not convincing. Suggest adding LLM-judge or human evaluation (or automatic structural/proof verifiers where possible) for long-form tasks to show actual quality gains or parity.

3. Only 3B instruction models are considered; results may not transfer to larger/production-grade models. Suggest adding at least one larger model (e.g., 7B/8B) to demonstrate scalability.

4. The failure of pure probability rewards on long answers is somewhat expected (vanishing probabilities). It would be useful to test simple fixes: 1) Log transform (e.g., log1p), likelihood ratio against a frozen reference model, or per-sample normalization to stabilize scales. 2) Entropy or variance-aware control variates beyond RLOO.

5. The gradient splits into a REINFORCE term plus a direct SFT-like term. An ablation that removes the direct SFT term would clarify whether the benefits stem from the policy gradient on CoT vs pure SFT on answers conditioned on CoTs.

**Questions:**

See weakness.

---

### Note · Authors · 2025-11-17

**Comment:**

We thank the reviewers for their insightful reviews. We are withdrawing this paper and will improve upon it, incorporating your remarks.

**Withdrawal Confirmation:**

I have read and agree with the venue's withdrawal policy on behalf of myself and my co-authors.